# Development of a Novel Biomarker for the Progression of Idiopathic Pulmonary Fibrosis

**DOI:** 10.3390/ijms25010599

**Published:** 2024-01-02

**Authors:** Hye Ju Yeo, Mihyang Ha, Dong Hoon Shin, Hye Rin Lee, Yun Hak Kim, Woo Hyun Cho

**Affiliations:** 1Department of Internal Medicine, School of Medicine, Pusan National University, Yangsan 50612, Republic of Korea; dugpwn@naver.com; 2Division of Pulmonary, Allergy, and Critical Care Medicine, Department of Internal Medicine, Pusan National University Yangsan Hospital, Yangsan 50612, Republic of Korea; 3Research Institute for Convergence of Biomedical Science and Technology, Pusan National University Yangsan Hospital, Yangsan 50612, Republic of Korea; donghshin@chol.com (D.H.S.); hrlee01070@gmail.com (H.R.L.); 4Interdisciplinary Program of Genomic Data Science, Pusan National University, Busan 46241, Republic of Korea; mh2059389@naver.com; 5Department of Nuclear Medicine, Pusan National University Medical Research Institute, Pusan National University Hospital, Busan 49241, Republic of Korea; 6Department of Pathology, School of Medicine, Pusan National University, Yangsan 50612, Republic of Korea; 7Department of Anatomy, School of Medicine, Pusan National University, Yangsan 50612, Republic of Korea; 8Department of Biomedical Informatics, School of Medicine, Pusan National University, Yangsan 50612, Republic of Korea

**Keywords:** IPF, prediction, progression, cathepsin, prognosis

## Abstract

The progression of idiopathic pulmonary fibrosis (IPF) is diverse and unpredictable. We identified and validated a new biomarker for IPF progression. To identify a candidate gene to predict progression, we assessed differentially expressed genes in patients with advanced IPF compared with early IPF and controls in three lung sample cohorts. Candidate gene expression was confirmed using immunohistochemistry and Western blotting of lung tissue samples from an independent IPF clinical cohort. Biomarker potential was assessed using an enzyme-linked immunosorbent assay of serum samples from the retrospective validation cohort. We verified that the final candidate gene reflected the progression of IPF in a prospective validation cohort. In the RNA-seq comparative analysis of lung tissues, *CD276*, *COL7A1*, *CTSB*, *GLI2*, *PIK3R2*, *PRAF2*, *IGF2BP3*, and *NUPR1* were up-regulated, and *ADAMTS8* was down-regulated in the samples of advanced IPF. Only CTSB showed significant differences in expression based on Western blotting (*n* = 12; *p* < 0.001) and immunohistochemistry between the three groups of the independent IPF cohort. In the retrospective validation cohort (*n* = 78), serum CTSB levels were higher in the progressive group (*n* = 25) than in the control (*n* = 29, mean 7.37 ng/mL vs. 2.70 ng/mL, *p* < 0.001) and nonprogressive groups (*n* = 24, mean 7.37 ng/mL vs. 2.56 ng/mL, *p* < 0.001). In the prospective validation cohort (*n* = 129), serum CTSB levels were higher in the progressive group than in the nonprogressive group (mean 8.30 ng/mL vs. 3.00 ng/mL, *p* < 0.001). After adjusting for baseline FVC, we found that CTSB was independently associated with IPF progression (adjusted OR = 2.61, *p* < 0.001). Serum CTSB levels significantly predicted IPF progression (AUC = 0.944, *p* < 0.001). Serum CTSB level significantly distinguished the progression of IPF from the non-progression of IPF or healthy control.

## 1. Introduction

Idiopathic pulmonary fibrosis (IPF) is a fatal, progressive, fibrotic lung disease with an estimated prevalence of approximately 0.5 to 27.9 per 100,000 [1]. Recently, two antifibrotic drugs were shown to have efficacy in decreasing the rate of decline in lung function in IPF [2,3]. Despite the availability of novel drugs, IPF is still associated with a high mortality rate, its clinical course is variable and unpredictable, and many IPF patients are untreated [4,5,6,7]. Almost one-third of IPF patients are not treated due to incorrect perceptions of “mild” or “stable” disease behavior [8]. In addition, prompt treatment with antifibrotic agents is hampered by many issues, such as a lack of confidence in IPF diagnosis, cost, and concerns regarding the side effects of these drugs [9]. Thus, a useful biomarker is required to convince physicians of the need for antifibrotic agents, determine therapeutic responses, and detect disease progression early.

Recently, research on new classifications of IPF phenotypes and individualized treatment strategies has been rapidly increasing [10,11,12,13]. Given the importance of rapid identification or understanding of disease progression in terms of treatment and prognosis, having molecular markers of disease progression is crucial. A recent study analyzing the transcriptome of RNA isolated from formalin-fixed paraffin-embedded (FFPE) provided new insights into the pathogenesis of IPF through a novel bioinformatics approach [14]. However, not much is known about the progression of IPF [15,16]. Traditionally, when wound healing pathways are impaired, activated fibroblasts continue to recruit and accumulate, forming fibrotic areas in the lung. This faulty wound-healing process can cause ongoing remodeling of IPF and damage the extracellular matrix (ECM), leading to the progression of IPF and end-stage lung disease [17]. Cysteine cathepsins are usually found in lysosomes, where they are involved in intracellular protein turnover and are currently considered to be critical players in ECM remodeling in vivo [18]. The role of cathepsin in IPF progression is not yet well known. To identify a candidate gene to reflect the progression of IPF, we analyzed the gene expression patterns in lung samples from patients with advanced IPF, those with early IPF, and controls. After specifying candidate genes from lung tissues of an independent IPF cohort, we retrospectively confirmed the association between biomarkers and IPF progression via blood samples from the retrospective test cohort. Finally, we evaluated whether the biomarker reflected IPF progression independent of clinical factors in the prospective validation cohort. The workflow of this study is illustrated in Figure 1.

## 2. Results

### 2.1. Identification of Potential Target Genes for IPF Progression

We first independently compared the different gene expression levels among a total of 78 lung tissue samples from control, early IPF, and advanced IPF patients in the GSE10667, GSE24206, and PNU datasets. We found common DEGs that were significantly up-regulated or down-regulated in the control, advanced IPF, and early IPF datasets (Figure 2A). As a result, we identified 13 up-regulated genes (*ALDH16A1*, *CD276*, *CLEC11A*, *COL7A1*, *CTSB*, *FDXR*, *GLI2*, *HOMER3*, *PIK3R2*, *PRAF2*, *SNED1*, *TCIRG1*, and *TYMS*) and three down-regulated genes (*ADAMTS8*, *GRIA1*, and *SGMS1*) (Figure 2B–D and Appendix A). Changes in the expression levels of the DEGs are shown in Appendix A and Appendix A.

### 2.2. Primary Verification of Candidate Gene Expression in Human IPF Lung Tissue

We searched the published literature, including the 16 genes common to the three datasets, and we prioritized nine genes based on published literature [19,20,21,22,23,24,25,26,27]. As a result, gene validation was conducted for only nine genes, including eight up-regulated genes (*CD276*, *COL7A1*, *CTSB*, *GLI2*, *PIK3R2*, *PRAF2*, *IGF2BP3*, and *NUPR1*) and one down-regulated gene (*ADAMTS8*).

Immunohistochemical staining was performed on lung tissue samples (control 3 vs. early IPF 6 vs. advanced IPF 6) to assess the tissue expression of candidate genes and their histological features. Among these, only cathepsin B (*CTSB*) expression was increased in the advanced IPF group compared to the control and early IPF groups, as revealed by the staining of macrophages in the honeycomb region (Figure 3A–F). A negative control was added as a Appendix A. Western blot analysis of lung lysates showed that *CTSB* was significantly up-regulated in the advanced IPF group compared to the control and early IPF groups (control 3 vs. early IPF 5 vs. advanced IPF 4, 1 vs. 31.2 vs. 204, *p* < 0.001, Figure 3G,H).

### 2.3. Evaluation of Serum CTSB as a Biomarker for the Progression of IPF in the Test Cohort

To test the possibility of using CTSB as a biomarker, serum CTSB was measured in the test cohort (*n* = 78) by ELISA. Serum CTSB levels were significantly higher in the progressive group than in the control (7.37 ng/mL vs. 2.70 ng/mL; *p* < 0.001) and nonprogressive groups (7.37 ng/mL vs. 2.56 ng/mL; *p* < 0.001). The levels of IL-6 (control vs. nonprogressive vs. progressive; 0.29 ng/mL vs. 0.37 ng/mL vs. 0.39 ng/mL; *p* < 0.001) and TGF-β (control vs. nonprogressive vs. progressive; 0.47 ng/mL vs. 0.64 ng/mL vs. 0.82 ng/mL; *p* < 0.001) were increased in the progressive group compared to the control group. Results of ELISA tests performed on serum for all candidate proteins and patient characteristics of the test cohort are in Appendix A and Appendix A. The progressive group showed different clinical phenotypes in terms of age, gender, lung function, rate of lung function decline, and GAP stage in the test cohort (Appendix A).

### 2.4. Validation in the Prospective Cohort

To validate the performance of the serum CTSB as a biomarker, baseline serum CTSB levels were measured in the independent IPF clinical cohort (validation cohort; *n* = 129). The baseline characteristics are presented in Appendix A. The progressive group showed different clinical phenotypes in terms of age, lung function, rate of lung function decline, and GAP stage in the validation cohort (Appendix A). There was a significant difference in serum CTSB levels between the two groups (*p* < 0.001, Figure 4A), and serum CTSB levels significantly distinguished the nonprogressive group from the progressive group (AUC 0.944, 95% CI 0.91–0.98, *p* < 0.001, Figure 4B). In addition, serum CTSB levels correlated with annual lung function decline (declined FVC per year, %) (r = 0.723, *p* < 0.001, Figure 4C). There were significant differences in CTSB levels between the progressive and nonprogressive groups in each GAP stage (Appendix A). In each GAP stage, the CTSB level of the progressive patients was significantly higher than that of the nonprogressive patients (I: 10.1 vs. 3.5, *p* < 0.001; II: 7.9 vs. 3.4, *p* < 0.001; III: 9.0 vs. 3.2, *p* < 0.001, Appendix A). Serum CTSB levels significantly distinguished the nonprogressive group from the progressive group in each subgroup (I: AUC = 0.977, *p* < 0.001; II: AUC = 0.927, *p* < 0.001; III: AUC = 0.952, *p* < 0.001; Appendix A).

### 2.5. Clinical Factors Associated with the Progression of IPF in Pooled Serum Samples

In the univariate logistic regression for overall cohorts (*n* = 178), age (OR = 0.94, 95% CI 0.90–0.98, *p* = 0.006), BMI (OR = 0.89, 95% CI 0.81–0.99, *p* = 0.024), baseline FVC (OR = 0.96, 95% CI 0.94–0.98, *p* < 0.001), and CTSB level (ng/mL) (OR = 2.57, 95% CI 1.95–3.38, *p* < 0.001) were significantly associated with progressive IPF (Figure 5A). We adjusted for important clinical factors, and the adjusted OR of CTSB is presented in Figure 5B. After we adjusted for the baseline FVC, we found that CTSB was significantly associated with progressive IPF (adjusted OR = 2.61, 95% CI 1.94–3.51, *p* < 0.001, Figure 5B).

## 3. Discussion

This study shows the possibility of using CTSB as a novel biomarker for reflecting the disease progression of IPF. We discovered blood biomarkers relevant to the progression of IPF and tested and validated them in each independent cohort. The serum CTSB level in progressive patients was significantly higher than that in controls and nonprogressive IPF patients. The baseline serum CTSB level was significantly related to the degree of lung function decline one year later and significantly predicted the progression of IPF. After adjusting for age, sex, BMI, and baseline lung function, serum CTSB levels were significantly associated with the progression of IPF. Regardless of the GAP stage, CTSB predicted progression well.

Developing biomarkers that reflect and predict the progression of IPF is crucial for optimizing individual treatment by identifying treatment needs and evaluating treatment response. Currently, progression is determined through lung function tests, high-resolution computed tomography, and clinical symptoms, which are incredibly subjective, biohazardous, and may vary depending on the clinician’s experience. In addition, due to the unpredictable clinical course of IPF, developing blood biomarkers that predict progression is an urgent and critical challenge. Several molecules involved in epithelial cell injury, matrix remodeling, and immune regulation have been proposed as promising biomarker candidates [28]. The TOLLIP rs5743890 C/T genotype was associated with rapid disease progression in 62 Caucasians with IPF, but additional validation was not performed [29]. KL-6 predicts the acute exacerbation risk of IPF but is not associated with mortality [30]. Only serial increases in KL-6 were associated with poor prognosis in 66 Japanese patients with IPF [31]. Recently, periostin predicted IPF progression compared to SP-D or KL-6 in 60 Japanese patients with IPF [32]. Although several candidates have shown promising results for disease progression, clinically effective and specific biomarkers for the progression of IPF are still lacking [33,34,35]. This study shows an unbiased approach to discovering blood biomarkers relevant to the progression of IPF. We used a statistical approach to narrow the potential target range of candidate genes and pooled the open-source RNA-seq datasets and the in-house RNA-seq dataset. Among the 16 genes common to the three datasets, we prioritized nine genes using published literature. After IHC and Western blotting, ELISA discovered the final candidate biomarker in the test cohort. Finally, the predictive power of CTSB for progressive IPF was confirmed in an independent validation cohort. We identified a more specific biomarker for IPF progression through the two-step verification process.

Traditionally, excessive ECM deposition is central to the pathogenesis of IPF. The production, deposition, and remodeling of collagen, a significant component of the ECM, is a dynamic process. In IPF, the balance between synthesis and degradation of these collagens and other ECM proteins is disrupted [36]. Physiologically, CTSB is integrated into almost all lysosome-related processes, including protein turnover, degradation, and some-mediated cell death [37]. CTSB can lead to the development of various pathological processes through the degradation and remodeling of the ECM. Additionally, CTSB initiates the proteolytic cascade by activating other tumor-promoting proteases, including matrix metalloproteinases and urokinase-type proenzyme activators. Under pathological conditions, CTSB can hydrolyze a variety of ECM components. However, the role of CTSB in IPF is not yet well understood. Although several recent studies have discovered signature genes for IPF, CTSB has still not received attention (Appendix A). Only one study showed that CTSB was among the top 50 differentially expressed genes between stable and progressive IPF [38]. Generally, CTSB contributes to the degradation of several types of collagen and elastin in the extracellular space [39,40]. Increased amounts and activity of cathepsins after bleomycin-induced lung injury are associated with lung fibrosis [41,42,43,44]. Recent studies have visualized the extent of pulmonary fibrosis in patients with IPF using a cathepsin-targeted imaging probe [45,46]. To our knowledge, despite its potential as a biomarker, no studies have evaluated the clinical usefulness of CTSB in assessing IPF progression. In this study, CTSB was found to be associated with progression regardless of the patient’s initial lung function, and it consistently reflected progression regardless of the GAP stage. Even after adjustment for various clinical factors, CTSB showed an independent association with progression. Patients with high serum CTSB may have a progressive phenotype of IPF, requiring understanding and careful clinical decisions regarding the timely initiation of antifibrotic agents or second-line treatment for suboptimal treatment response. Further basic and clinical research involving a large IPF cohort is needed to extend our understanding of the role of CTSB in the progression of IPF and to confirm the performance of CTSB as a biomarker.

Our study has several strengths, including connecting alterations in protein levels to gene expression and offering hypotheses on the differential pathogenesis related to subsets of patients with IPF; however, it also has some limitations. First, the number of in-house lung tissue samples used for RNA sequencing was small. A pooled analysis was performed using a genetic dataset from external cohorts to overcome this limitation. We used the most relevant method to determine the specific gene signature of IPF progression [47,48]. Second, there may be selection bias given the retrospectively assessed patient group from a single center, despite some prospective sample collection. However, we conducted a two-step verification of serum CTSB levels in independent cohorts. In particular, this study has great value in that it generates hypotheses and presents experimental validation through an unbiased approach. We linked the gene expression data to histopathological findings and validated our findings in a longitudinal clinical cohort. CTSB offers the potential to reflect the progression of IPF. Our results suggest a novel biomarker for identifying patients with IPF progression. Further research is warranted for multicenter international cohorts. On the one hand, it is necessary to elucidate the primary mechanism of CTSB using a murine model of IPF and a cell model of type II alveolar epithelium.

## 4. Materials and Methods

### 4.1. Study Population and Sample Processing

We collected in-house lung tissue samples from lung transplant recipients with advanced IPF, samples from early IPF patients obtained after surgical lung biopsy, and samples from healthy donor lungs obtained after lung volume reduction for the size matching in the Pusan National University Yangsan Hospital (PNUYH) biobank with informed consent. We also collected serum from IPF patients attending outpatient clinics at the PNUYH Biobank with informed consent. Lung tissue samples and serum were stored at −80 °C. Lung tissue samples for IHC and Western blot analysis and serum samples for ELISA were obtained from the PNUYH biobank. RNA sequencing was performed on paraffin blocks obtained after surgical lung biopsy of early IPF lungs of patients (*n* = 3), from explanted native lungs (advanced, *n* = 3) of lung transplant recipients, and from normal control subjects (donor lungs, *n* = 3). RNA extraction, library construction, and sequencing details are presented in Appendix A. The diagnosis of IPF was based on the definitions of the American Thoracic Society and European Respiratory Society [49]. Expert pulmonologists and pathologists reviewed all cases. All studies were approved by the Institutional Review Board of Pusan National University Yangsan Hospital (PNUYH IRB No 04-2021-016). All tissue and serum from the biobank were anonymized, and the need for informed consent was waived. All methods were performed in accordance with the relevant guidelines and regulations.

### 4.2. Development of a Potential Biomarker

#### 4.2.1. Differentially Expressed Genes

We downloaded the gene expression profiling data of the IPF array from the Gene Expression Omnibus (GEO) database (http://www.ncbi.nlm.nih.gov/geo/ (accessed on 31 December 2023)) using the R package “GEOquery (v 2.52.0)”. Accession numbers were GSE10667 and GSE24206, which were annotated using the Agilent-014850 Whole Human Genome Microarray 4 × 44 K G4112F and Affymetrix Human Genome U133 Plus 2.0 Array platforms, respectively. We analyzed differentially expressed genes (DEGs) between the normal lung tissues, acutely exacerbated or advanced IPF lung tissues, and early IPF lung tissue in each dataset and in-house RNA-seq dataset (PNU). Then, we extracted the common genes from the 3 datasets.

#### 4.2.2. Immunohistochemistry and Western Blot

To verify the DEGs, candidate proteins were confirmed via IHC staining and Western blotting in lung tissue samples from an independent in-house IPF tissue cohort. IHC staining was conducted in control 3 vs. early IPF 6 vs. advanced IPF 6, and Western blotting was conducted in control 3 vs. early IPF 5 vs. advanced IPF 4. IHC staining for candidate proteins, hematoxylin–eosin staining (HE), and Western blotting were performed using antibodies as described in Appendix A. ELISA and IHC details are presented in Appendix A.

### 4.3. Retrospective and Prospective Validation of the Developed Biomarker

Serum samples from two cohorts were used for biomarker validation. The IPF blood cohort was classified into nonprogressive and progressive groups, which were divided by the retrospective presence of lung function decline as follows. We calculated the decline in FVC using the relative change: (baseline FVC (%) − FVC (%) at 1 year)/baseline FVC (%) × 100 [50]. The progressive group was assigned if there was a ≥ 10% relative decline in FVC from baseline over 1 year (e.g., from 60% predicted to 54% predicted) [38,51,52]. The others were assigned to the nonprogressive group. In the retrospective cohort, which had evidence of prior lung function decline, the potential biomarker was verified based on serum samples through ELISA: progressive (*n* = 25), nonprogressive (*n* = 24), and control groups (*n* = 29). We evaluated proinflammatory cytokines and profibrotic growth factors associated with IPF, along with the genes of interest [53,54]. These were determined using commercially available ELISA kits, as described in Appendix A. Detailed clinical information regarding the subjects is summarized in Appendix A. Finally, we evaluated the candidate protein as a biomarker in the independent prospective validation cohort (*n* = 129). We assessed the candidate protein’s correlation with and predictive power regarding lung function decline to confirm its monitoring and predictive performance as a biomarker. In addition, risk factors for IPF progression were analyzed.

### 4.4. Statistical Analysis

To compare the different expression levels of genes in the three groups, the groups were independently evaluated using ANOVA (analysis of variance) or the Kruskal–Wallis rank sum test. Once significant differences were found, the Scheffe post hoc test in the R package “Desctools (v 0.99.39)” or Bonferroni’s multiple comparison tests in the R package “dunn.test (v 1.3.5)” was performed. Statistical significance was set at a *p* value < 0.05. The results are presented as the mean ± standard deviation (SD) or median ± interquartile range (IQR), depending on the statistical method. Then, we identified the common DEGs based on statistical significance in the GSE10667, GSE24206, and PNU datasets. The R packages “ggvenn (v 0.1.8)” and “gplots (v 3.0.4)” were used to draw the plots, and all statistical analyses were performed using R software version 3.6.3. To compare the quantitative variables of the progressive group and nonprogressive group in the test and validation cohorts, we used the parametric and non-parametric methods as appropriate. For example, we used the independent *t*-test (parametric method) or Mann–Whitney U test (non-parametric method) to compare means of two groups, and ANOVA (parametric method) or Kruskal–Wallis rank sum test (non-parametric method) to compare means of three groups. In addition, the chi-square test or Fisher’s exact test was used to compare qualitative variables based on whether the data were normally distributed. We assessed the association between CTSB (ng/mL) and progressive IPF using a binary logistic regression analysis and reported the odds ratios (ORs) and *p* values. Furthermore, major clinical factors were included in the logistic regression model as covariates and expressed as adjusted ORs. Finally, receiver operating characteristic (ROC) curves were calculated to assess the overall diagnostic function of the CTSB levels.

## Figures and Tables

**Figure 1 ijms-25-00599-f001:**
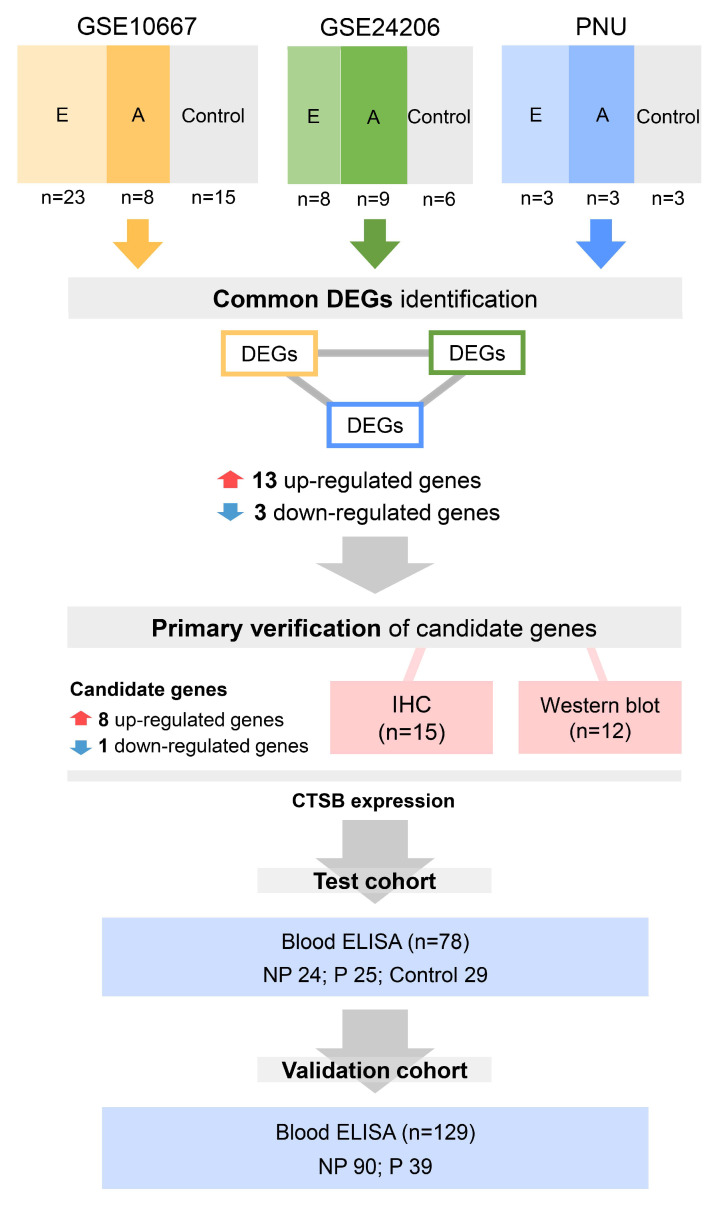
Workflow of this study. We first independently compared the different gene expression levels among a total of 78 lung tissue samples from control, early IPF, and advanced IPF patients in the GSE10667, GSE24206, and PNU datasets. We found common DEGs that were significantly 13 up-regulated or three down-regulated in the control, advanced IPF, and early IPF datasets. Among the 16 genes common to the three datasets, gene validation was conducted for only nine genes, including eight up-regulated genes (CD276, COL7A1, CTSB, GLI2, PIK3R2, PRAF2, IGF2BP3, and NUPR1) and one down-regulated gene (ADAMTS8). Immunohistochemical staining was performed on 15 lung tissue samples (control three vs. early IPF 6 vs. advanced IPF 6). Western blot analysis was also performed on 12 lung tissue samples (control three vs. early IPF 5 vs. advanced IPF 4). To test the possibility of using CTSB as a biomarker, serum CTSB was retrospectively measured in the test cohort (*n* = 78) using ELISA. Finally, to validate the possibility of using CTSB as a biomarker, serum CTSB levels were prospectively measured in the independent IPF clinical cohort (validation cohort; *n* = 129). E: early IPF; A: advanced IPF; DEGs: differentially expressed genes; IHC: immunohistochemistry; ELISA: enzyme-linked immunosorbent assay; NP: nonprogressive IPF; P: progressive IPF; IPF: idiopathic pulmonary fibrosis.

**Figure 2 ijms-25-00599-f002:**
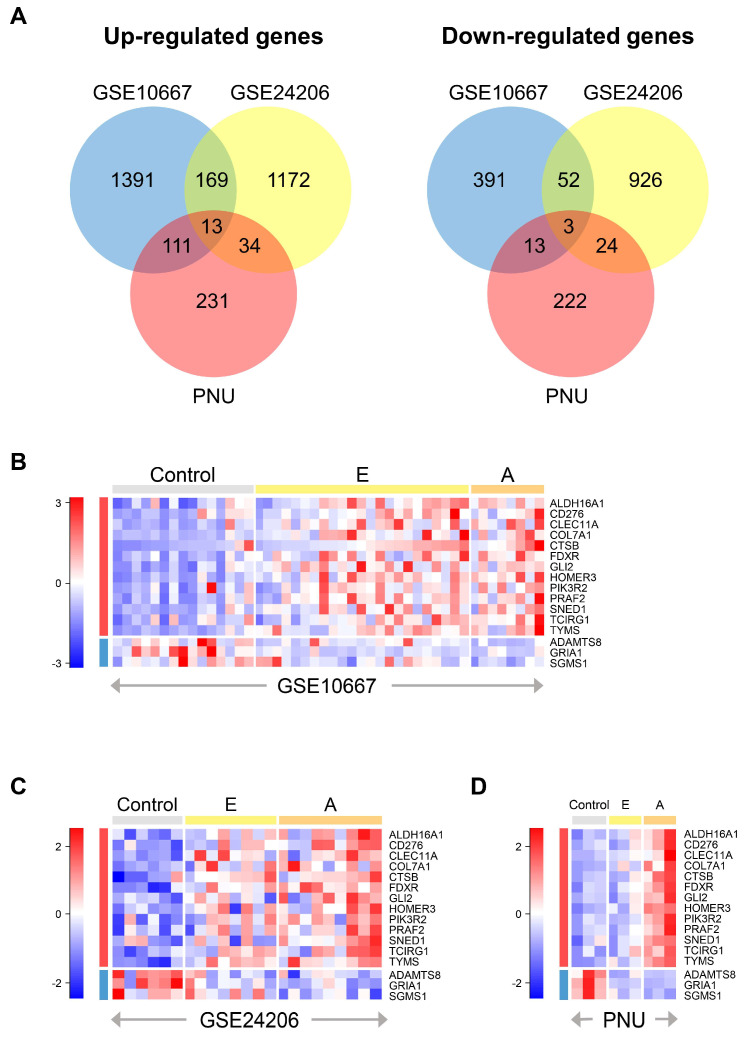
Identification of DEGs. (**A**) Venn diagrams show the overlapping DEGs among the three datasets: up-regulated genes (left) and down-regulated genes (right). We used the ANOVA or Kruskal–Wallis rank sum test to compare different expression levels among the three groups (control, early IPF, and advanced IPF). Each circle represents the number of independently up or down-regulated genes in each dataset. Heatmap of scaled gene expression of 16 common DEGs in the three datasets: (**B**) GSE10667, (**C**) GSE24206, and (**D**) PNU. Red indicates high relative expression, and blue indicates low relative expression. The rows represent 13 up-regulated genes and three down-regulated genes in order, and the gray, yellow, and orange bars at the top of the heatmap represent control, early, and advanced IPF samples, respectively. DEGs: differentially expressed genes; E: early IPF; A: advanced IPF; DEGs: differentially expressed genes; IPF: idiopathic pulmonary fibrosis; ANOVA: analysis of variance.

**Figure 3 ijms-25-00599-f003:**
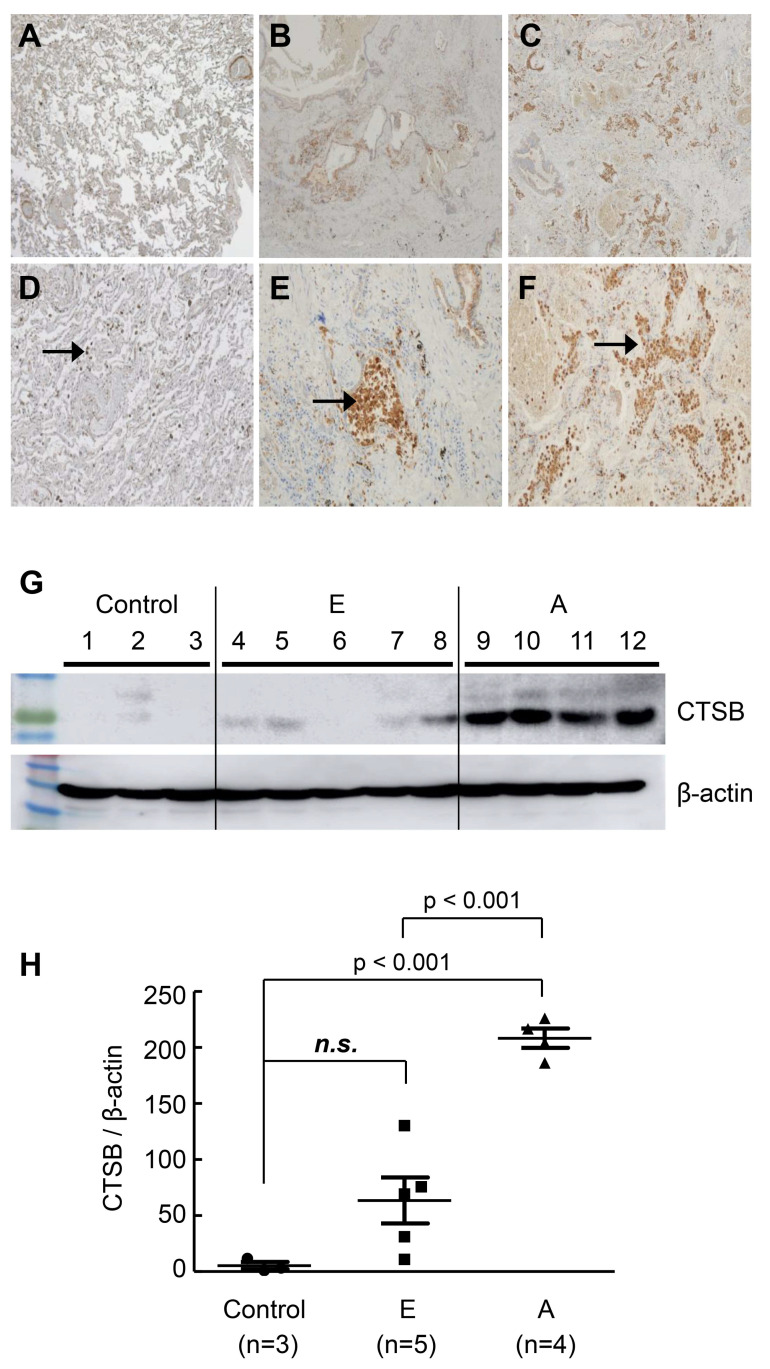
Representative image of immunohistochemistry CTSB and Western blot. (**A**) Control; (**B**) early IPF; (**C**) advanced IPF; (**D**) control; (**E**) early IPF; (**F**) advanced IPF; IHC staining and Western blot were performed on tissue from the right lower lobe area of control, early IPF, and advanced IPF. The staining of CTSB was increased in advanced IPF compared to that in the control and in early IPF, marking macrophages in the honeycomb region. Arrows indicate macrophages stained for CTSB. A negative control was added, as shown in Appendix A. (**G**) Western blot. In the Western blot, each group was marked separately using a dividing line. (**H**) CTSB expression levels in control vs. early IPF vs. advanced IPF. CTSB was increased in advanced IPF compared to the control and early IPF. Each short bar below and above represents the first and third quartiles of expression, and each central long bar represents the median value. E: early IPF; A: advanced IPF; IPF: idiopathic pulmonary fibrosis; n.s.: no significance.

**Figure 4 ijms-25-00599-f004:**
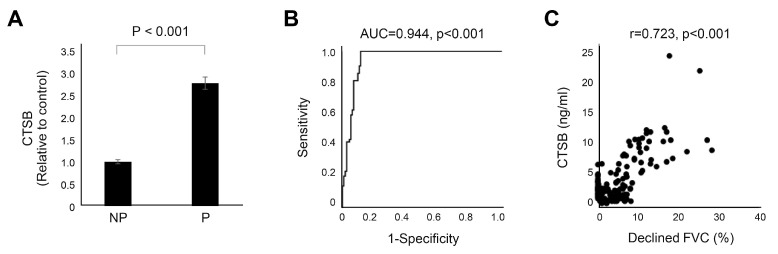
CTSB in progressive IPF. Data represent normalized values relative to the control, which is considered to have a value of 1. (**A**) Bar charts with error bars are presented as the median and 95% confidence intervals around the median. (**B**) ROC curve of CTSB. CTSB significantly predicted progressive IPF (AUC = 0.944, *p* < 0.001). (**C**) There was a significant correlation between CTSB and decreased FVC per year (r = 0.723, *p* < 0.001). NP: nonprogressive IPF; P: progressive IPF; AUC: area under the curve; FVC: forced vital capacity; ROC: receiver operating characteristic.

**Figure 5 ijms-25-00599-f005:**
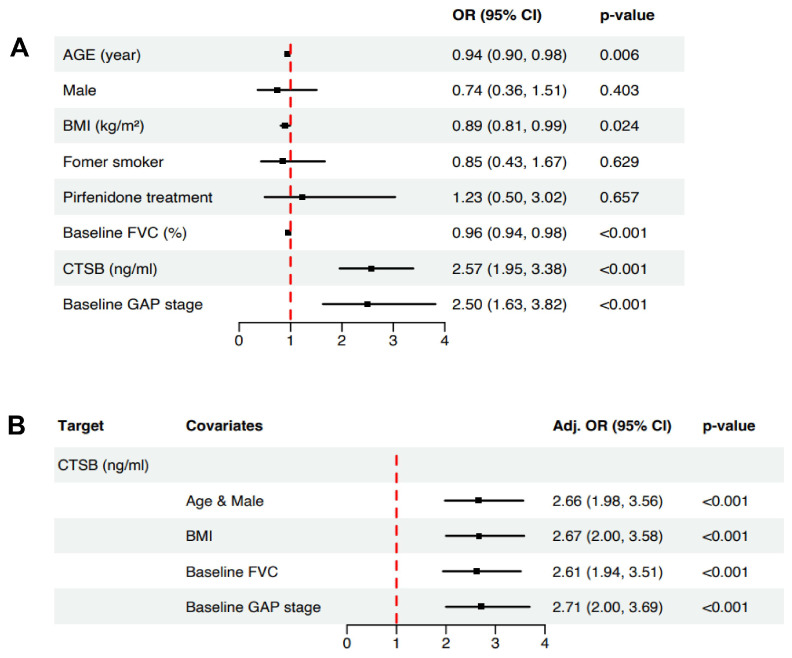
Logistic regression analysis. (**A**) Forest plot for univariate logistic regression analysis. As a result, age (OR = 0.94, 95% CI 0.90–0.98, *p* = 0.006), BMI (OR = 0.89, 95% CI 0.81–0.99, *p* = 0.024), baseline FVC (OR = 0.96, 95% CI 0.94–0.98, *p* < 0.001), and CTSB level (OR = 2.57, 95% CI 1.95–3.38, *p* < 0.001) were significantly associated with progressive IPF. (**B**) Forest plot for adjusted OR analysis. After we adjusted for baseline FVC, CTSB was significantly associated with progressive IPF (adjusted OR = 2.61, 95% CI 1.94–3.51, *p* < 0.001). OR: odds ratio; CI: confidence interval; FVC: forced vital capacity; GAP: gender-age-physiology.

## Data Availability

All data generated or analyzed during this study are included in this published article and its Appendix A. We downloaded the gene expression profiling data of the IPF array from the Gene Expression Omnibus (GEO) database (http://www.ncbi.nlm.nih.gov/geo/ accessed on 31 December 2023) using the R package “GEOquery (v 2.52.0)”. Accession numbers were GSE10667 and GSE24206.

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
