# Peer review of "Development of a Novel Biomarker for the Progression of Idiopathic Pulmonary Fibrosis"

_ijms, 2024, doi:10.3390/ijms25010599_

Round 1

Reviewer 1 Report

Comments and Suggestions for Authors

The manuscript evaluates a novel biomarker to identify progressive IPF using an in house cohort and analysing two public gene expression datasets for IPF. The text is well written and easy to follow. However, the figure legend could be improved to make it easier to understand the figure without having to read the full text.

1.     Background could be extended with more information on different states of IPF. Introduce remodeling as CTSB is important for that.

2.     Figure 1 needs more explanation in the legend. What type of material, what the cohorts are, what DEG is etc.

3.     The Methods sections is too brief regarding RNA seq on the in house PNU cohort. Describe how the samples were processed and analyzed. State that you find method description on ELISA and IHC in the supplement.

4.     Clarify in Results 3.1 that DEG are identified in tissue samples

5.     Figure 2. Write out abbreviations. Include statistics used to define DEG

6.     Figure 3. Which part of the lung are the biopsies? Include negative controls in the figure or in supplement. IHC in higher magnification to evaluate the specificity of the staining. Is the staining intracellular? Are only macrophages stained? Indicate structural features. In figure legend it says early and advanced IPF and in figure non-progressive and progressive. In other figures early and advanced is used. Be consistent or explain the difference.

7.     Figure 4. Inconsistent with Figure 3 in defining groups.

8.     Supplementary Figure 2. State what method (ELISA?) and material (tissue, serum?) used in the analysis.

9.     Discussion. A bit brief. Could go more into detail of the role of CTSB.

Reviewer 2 Report

Comments and Suggestions for Authors

The paper's exploration of CTSB as a potential biomarker for monitoring IPF progression is a commendable effort in an area that desperately needs improved diagnostic and prognostic tools. However, several weaknesses and limitations in the methodology and interpretation of the results need to be addressed.

- Sample size and bias:
The study's reliance on a small sample size, particularly the in-house lung tissue samples for RNA sequencing, raises concerns about its representativeness. In addition, the retrospective assessment of patients from a single centre introduces potential selection bias, which affects the generalisability of the findings. Although attempts were made to validate the results in independent cohorts, the initial small sample size may limit the robustness of the conclusions drawn.

- Biomarker validation:
The paper's transition from gene expression analysis to identification of CTSB as a biomarker for IPF progression lacks comprehensive validation in larger, more diverse cohorts. Although the study demonstrated higher CTSB levels in the progressive group, more extensive validation in a larger, multi-centre cohort would strengthen the credibility of CTSB as a reliable biomarker. In addition, further validation using longitudinal data would strengthen the predictive power of CTSB for tracking disease progression over time.

- Interpretation and Clinical Application:

While the study provides promising data on CTSB, caution should be exercised in translating these findings into clinical practice. The paper acknowledges the need for further research to understand the role of CTSB in IPF progression. There's a gap in explaining the specific mechanistic involvement of CTSB in IPF pathology, which is crucial for its clinical relevance and potential therapeutic targeting.

- Lack of comprehensive analysis:
The study focuses on CTSB without discussing potential confounding variables or other biomarkers that may influence IPF progression. Inclusion of a broader range of potential biomarkers and consideration of multifactorial aspects of disease progression could provide a more comprehensive understanding and potentially identify a panel of biomarkers for improved diagnostic accuracy.

- Methodological limitations:
The paper acknowledges its limitations in terms of sample size and selection bias. However, the strength of the study lies in its unbiased approach. Nonetheless, the authors could provide clearer rationale and strategies for overcoming these limitations, potentially suggesting avenues for future research to address these issues more effectively.

In conclusion the discovery of CTSB, as a biomarker for IPF progression is certainly promising. However it's important to acknowledge the limitations of this study, which currently prevent us from applying these findings in practice. It is crucial to conduct validation in larger and more diverse groups of patients and also gain a deeper understanding of the underlying molecular mechanisms that involve CTSB in IPF development. Only after these steps can we confidently consider incorporating CTSB as a biomarker for IPF. While this paper lays the groundwork, for investigations it still requires validation and refinement before being widely accepted as an IPF biomarker.

For these reason i suggest to reconsider the paper afeter major revision.

Reviewer 3 Report

Comments and Suggestions for Authors

In this study, Hye Ju Yeo et al. presents an intriguing exploratory study that aims to identify novel biomarkers for the progression of idiopathic pulmonary fibrosis (IPF). The authors have identified CTSB as a potential differentially expressed gene that could serve as a biomarker for IPF progression. While the research premise is promising, there are several critical aspects that need addressing to strengthen the findings:

1 Cohort Size and Validation Concerns:

The discovery cohort comprising three early-stage IPF patients, three advanced-stage IPF patients, and three controls appears insufficient to establish robust conclusions. While the authors have attempted experimental validation, the small sample size considerably undermines the reliability of the findings, and make readers to hardly believe the resutls. A larger cohort is essential to ensure the statistical power and reproducibility of the results.

2 Integration of Datasets and Literature Context:

The manuscript also references additional datasets (GSE10667 and GSE24206) derived from gene expression microarrays. The methodological approach to integrating RNA-Seq with microarray data lacks clarity. Besides, these databases have been extensively utilized in IPF research over the past decade, and there are more than 100 papers using these databases. It is suspicious why these studies not find CTSB as a potential biomarker for IPF progression? The authors need provide a detailed review table summarizing previous findings using these datasets and contrasting them with the current study's results, particularly regarding the newly reported role of CTSB.

3. Literature Sources for Gene Prioritization:

In lines 157-158, the authors prioritize nine genes from sixteen based on the literature but fail to cite specific sources. It is crucial to reference the literature that guided the prioritization to enable readers to follow the rationale and to verify the relevance of the selected genes.

4. Cohort Source and Sequencing Decisions:

The source of the test cohort (n=68) and the prospective cohort (n=129) is not disclosed. If these cohorts were obtained from Pusan National University Yangsan Hospital, it is unclear why sequencing was not performed on these additional samples. Clarification on the origin of these cohorts and the reasoning behind the methodological choices would add substantial value to the study's context and conclusions.

Comments on the Quality of English Language

Extensive editing of English language required

Round 2

Reviewer 2 Report

Comments and Suggestions for Authors

On the basis of the authors' responses, my recommendation is acceptance of the manuscript in its present form.

Reviewer 3 Report

Comments and Suggestions for Authors

No further comments.

Comments on the Quality of English Language

 Moderate editing of English language required